# The Classification of Rice Blast Resistant Seed Based on Ranman Spectroscopy and SVM

**DOI:** 10.3390/molecules27134091

**Published:** 2022-06-25

**Authors:** Yan He, Wei Zhang, Yongcai Ma, Jinyang Li, Bo Ma

**Affiliations:** 1Engineering College, Heilongjiang Bayi Agricultural University, Daqing 163319, China; heyanbyau@163.com (Y.H.); myc1631@163.com (Y.M.); ljy970118@163.com (J.L.); 2Qiqihar Branch of Heilongjiang Academy of Agricultural Sciences, Qiqihar 161006, China; mabo8210@haas.cn

**Keywords:** ranman spectroscopy, rice blast, resistant varieties, optimize support vector machine algorithm, artificial bee colony algorithm

## Abstract

Rice blast is a serious threat to rice yield. Breeding disease-resistant varieties is one of the most economical and effective ways to prevent damage from rice blast. The traditional identification of resistant rice seeds has some shortcoming, such as long possession time, high cost and complex operation. The purpose of this study was to develop an optimal prediction model for determining resistant rice seeds using Ranman spectroscopy. First, the support vector machine (SVM), BP neural network (BP) and probabilistic neural network (PNN) models were initially established on the original spectral data. Second, due to the recognition accuracy of the Raw-SVM model, the running time was fast. The support vector machine model was selected for optimization, and four improved support vector machine models (ABC-SVM (artificial bee colony algorithm, ABC), IABC-SVM (improving the artificial bee colony algorithm, IABC), GSA-SVM (gravity search algorithm, GSA) and GWO-SVM (gray wolf algorithm, GWO)) were used to identify resistant rice seeds. The difference in modeling accuracy and running time between the improved support vector machine model established in feature wavelengths and full wavelengths (200–3202 cm^−1^) was compared. Finally, five spectral preproccessing algorithms, Savitzky–Golay 1-Der (SGD), Savitzky–Golay Smoothing (SGS), baseline (Base), multivariate scatter correction (MSC) and standard normal variable (SNV), were used to preprocess the original spectra. The random forest algorithm (RF) was used to extract the characteristic wavelengths. After different spectral preproccessing algorithms and the RF feature extraction, the improved support vector machine models were established. The results show that the recognition accuracy of the optimal IABC-SVM model based on the original data was 71%. Among the five spectral preproccessing algorithms, the SNV algorithm’s accuracy was the best. The accuracy of the test set in the IABC-SVM model was 100%, and the running time was 13 s. After SNV algorithms and the RF feature extraction, the classification accuracy of the IABC-SVM model did not decrease, and the running time was shortened to 9 s. This demonstrates the feasibility and effectiveness of IABC in SVM parameter optimization, with higher prediction accuracy and better stability. Therefore, the improved support vector machine model based on Ranman spectroscopy can be applied to the fast and non-destructive identification of resistant rice seeds.

## 1. Introduction

Rice is one of the most important grain crops in the world, feeding more than half of the world’s people, and its yield stability is crucial to guarantee social stability and economic development [1]. Rice blast, a fungal disease that threatens rice production [2], is known as the “cancer” of rice. It occurs in the whole growing period of rice and can lead to a yield reduction of 10–35% or even no harvest. Cultivating rice varieties with resistance to the disease is the most cost-effective way to prevent rice blast [3], which plays an important role in ensuring grain security.

Identification of rice seeds has always been an important issue in crop breeding [4]. Traditional rice seed detection methods include morphological identification, microscopic identification, simple sequence repeat (SSR) molecular markers and the field experiment method [5]. The field experiment method method has defects such as unreliable resistance evaluation, long detection time and other environmental limiting factors (wind speed, temperature and humidity). The SSR molecular marker method has high accuracy for the detection of rice varieties [6] but requires specialized laboratories with more expensive equipment and demanding personnel technology. Therefore, this study tries to propose a fast and accurate identification method for resistant rice seeds.

Ranman spectroscopy has the advantages of convenient operation, high sensitivity and good reproducibility, which can effectively overcome many of the shortcomings of traditional detection methods [7]. The aim is to describe difference in the biological mechanisms of rice seeds using the characteristics of Ranman spectroscopy [8]. The Ranman spectra of different-resistance rice seeds were collected to classify the rice seeds with different resistance according to the spectral features of different molecular structures inside the seeds. Then, the original Ranman spectral data obtained were preprocessed to remove noise nuisances caused by external light, temperature and instruments [9]. Analyzing the differences in Ranman spectral curves caused by the internal protein, nucleic acid, carbohydrate and so on of differently resistant rice seed allows the realization of the rapid and non-destructive detection of differently resistant rice seeds. In recent years, support vector machines (SVM) has also been studied and applied in the field of spectral analysis. Scholars have carried out a lot of studies on rice varieties, such as japonica [10], Vietnam rice [11], cold-region rice [12] and adulterated rice [13] using Ranman spectroscopy with SVM. In terms of rice blast detection, they are all about the identification of rice leaves [14] and plant images [15].

Therefore, this study provides a non-destructive method suitable for the high-generation screening of small batches of seeds using Ranman spectroscopy with SVM based on the detection of resistance to rice blast in rice seed.

## 2. Materials and Methods

### 2.1. Instrument and Equipment

A Advantage 532 desktop Ranman spectrometer produced by DeltaNu Company in the United States was used in the experiment, with a resolution of 1.4 cm^−1^ and a measurement range of 200–3400 cm^−1^, combined with ProScope HR software to obtain samples’ spectral curves. The preprocessing of the spectral data was implemented in The Unscrambler X 10.3 (64-bit) software, and the characteristic wavelengths’ extraction, modeling analysis and graph drawing were implemented in MATLAB R2016a.

### 2.2. Ranman Spectra Collection

The resistant rice seeds used in this research came from Heilongjiang Beifeng Agricultural Means of Production Group. The four kinds of rice-blast-resistant seeds were: high-resistance Longjing 33, high-susceptibility Longjing 56, high-resistance Longjing 34 and sensitive Longjing 36, which are all oval. Because the seeds were wrapped in chaff, the Ranman spectrometer could not penetrate the chaff to directly collect the spectral information of the resistant rice seeds. The outer chaff was manually removed from the rice seeds, and rice samples were obtained for later use. From each resistant rice variety, 60 grains were selected as the sample set (a total of 240 grains). For each sample, the spectrum measurement was repeated 3 times for each sample, and then the average spectrum was used as a representative spectrum for each sample. A total of 240 Ranman spectra of resistant rice were obtained.

### 2.3. Spectra Preprocessing and Feature Extraction

Because of some background noise in the raw spectral data, proper spectral preprocessing methods were considered to enhance the spectral features. In this paper, five spectral preprocessing methods are shown in Figure 1: Savitzky–Golay 1-Der (SGD), Savitzky–Golay smoothing (SGS), baseline (Base), multivariate scatter correction (MSC) and standard normal variable (SNV). It can be clearly seen from the preprocessing results that the spectral curve of the rice seeds were very similar, in the range of 200–3200 cm^−1^, and that the results of the SGD method are the most unsatisfactory. Therefore, it is impossible to directly distinguish differently resistant rice seeds from the spectral curve, and it is necessary to extract the characteristic wavelengths for further discriminant analysis.

Extracting characteristic variables from the full bands can reduce data redundancy and multicollinearity to a certain extent [16] and can improve the accuracy and reduce the running time of the resistant-rice-classification models. Random forest (RF) is a popular and very efficient algorithm, based on model-aggregation ideas, for both feature extraction and classification problems, introduced by Breiman [17]. The principle is to combine many binary decision trees built using several bootstrap samples coming from the learning sample L (Ranman spectral data), and choosing randomly at each node a subset of explanatory variables X (feature variables). First, the threshold value was set to 0.45, and the characteristic variables of the full wavelengths ere extracted using RF. The extracted 90 characteristic variables are shown in Figure 2. Then, five kinds of preprocessed data were used to extract feature variables by RF. After SNV, SGD, MSC, Base and SGS, 61, 30, 46, 46 and 46 feature variables were extracted by RF, respectively.

### 2.4. Discriminant Methods

#### 2.4.1. Support Vector Machine

Support vector machine (SVM) is based upon the principle of structural risk minimization, with salient properties of ease in generalization and fewer required training samples [18]. An SVM displayed substantial benefits when compared to other classification approaches. It is challenging to construct a linear classifier to separate the classes of data. In SVM, the transfer function is introduced to map the input vectors a high-dimensional space (generally a Hilbert space), which can effectively reduce the optimization complexity and improve the generalization capability. It then constructs a linear classification decision to classify the input spectral data with a maximum margin hyperplane. An SVM has also been found to be more effective and faster than other machine learning methods.

The computational parameters of the SVM model can be obtained by solving the following convex optimization problem with a ε-insensitivity loss function:(1)$$\min\frac{1}{2}\omega^{T}\omega +C\overset{N}{\underset{i=1}{\sum }}\xi_{i}\;s.t.{\;y}_{i}\left(\omega^{T}x_{i}+b\right)\geq 1-\xi_{i},\;\xi_{i}\geq 0,i=1,2,\cdots ,N$$

In general, the model in Equation (1) can be addressed by constructing a primal optimization problem based on a Lagrange function, which is given as below:(2)$$L\left(\omega ,b,\lambda \right)=\frac{1}{2}\omega^{T}\omega +C\overset{N}{\underset{i=1}{\sum }}\xi_{i}-\overset{N}{\underset{i=1}{\sum }}\lambda_{i}\left[y_{i}\left(\omega^{T}x_{i}+b\right)-1+\xi_{i}\right]-\overset{N}{\underset{i=1}{\sum }}\alpha_{i}\xi_{i}$$

Among,${\;\lambda }_{i}\geq 0,\;i=1,2,\cdots ,N$.

The dual function of linear indivisible problems is as below:(3)$$\mathrm{maxQ}\left(\lambda \right)=\overset{N}{\underset{i=1}{\sum }}\lambda_{i}-\frac{1}{2}\overset{N}{\underset{i,j=1}{\sum }}y_{i}y_{j}\lambda_{i}\lambda_{j}\left(\Phi_{x_{i}}\times \Phi_{x_{j}}\right)\;s.t.\;\overset{N}{\underset{j=1}{\sum }}y_{i}\lambda_{i}=0.1\lambda_{i}\leq C\;$$

Then, the decision function for the SVM model can be described as below:(4)$$f\left(x,\lambda \right)=\mathrm{sgn}\left(\sum_{\mathrm{SV}}y_{i}\lambda_{i}K\left(x_{i},x_{j}\right)+b\right)$$

Three pattern-recognition methods (support vector machine (SVM), BP neural network (BP) and probabilistic neural network (PNN)) were applied to the building of rice-classification models. The results of resistant-rice-classification models built by different modeling methods are shown in Table 1.

As can be seen, the optimum raw BP model (with classification accuracy of 97% and a running time of 487 s) was obtained using the raw spectra. The Raw-SVM model runs very fast, with an accuracy of 45%. If the raw BP model is optimized, the running time will be lengthier. With the research and development of swarm intelligence algorithms, many intelligent algorithms have been applied to the parameter optimization of SVM. In order to improve the classification results, the Raw-SVM model should be chosen for optimization and the optimal (c,g) parameters combination should be sought.

#### 2.4.2. Optimize Support Vector Machine

In recent years, the swarm intelligence algorithm has been widely used to optimize SVM parameters [19]. The artificial bee colony algorithm (ABC) is a new swarm intelligence algorithm that was proposed in 2005 by Karaboga [20]. In order to find an optimal solution through iterations, ABC calculates the evaluation value of a food source by formula. The ABC algorithm can converge on the global optimum faster, thus improving the accuracy of SVM in classification and accelerating the convergence speed of (c,g) parameter optimization. However, the traditional artificial bee colony algorithm easily falls into local extreme points in the later stage. Many scholars take advantages of the ABC algorithm to optimize SVM parameters. Luo et al. [21] introduced a chaotic sequence to re-initialize hireling bees. Zhou et al. [22] used stepwise optimization to transform the selection strategy. Kuang et al. [23] generated a chaotic sequence based on the local optimal solution and selected the optimal solution from the sequence as the new honey source location; Liu Xia et al. [24] used chaotic mapping to initialize the population. Liu et al. [25] optimized the ABC algorithm using a random dynamic local search operator. These optimizing ABC algorithms improved the search performance to a certain extent and at the same time fully verified the chaotic optimization algorithm’s advantages of being insensitive to initial values and demonstrating strong ergodicity.

In this paper, the ergodicity of the chaotic search algorithm (CS) is utilized. In the iterative process of the artificial bee colony algorithm, when the number of searches is greater than the set maximum number and a better nectar source has not been obtained, it will fall into the problem of local optimal solutions. The chaotic search algorithm [26] is introduced to generate chaotic sequences to form an improved artificial bee colony algorithm (IABC) based on the chaotic update strategy. The IABC algorithm has a chaotic update strategy scout bee, which traverses the chaotic sequence and compares the corresponding fitness values with those of the stagnant solution to find a better solution to replace the stagnant solution, so that the algorithm jumps out of the local optimum. The experimental results show that the artificial bee colony algorithm based on the chaotic-update strategy has accelerated the convergence speed, enhanced the later local jump-out ability and improved the SVM classification performance.

First, the bee colony is initialized. According to the results of many experiments, the value ranges of the nuclear parameter and the penalty factor C are determined to be (0, 0.01) and (1, 100), respectively. Using two-dimensional uniform design method, the value range of C is evenly divided into 25 squares. The range of the initial food source is represented by each square. Once the bee leaves the local optimal solution, it can find the square without the optimal solution among all the squares and generate the optimal solution among the remaining squares.

Second, the food sources are updated. The penalty factor C and the kernel function parameter γ of SVM are both optimized. The Euclidean distance between food source $\left(C_{1},\gamma_{1}\right)$ and food source $\left(C_{2},{\;\gamma }_{2}\right)$ can be expressed as:(5)$$d=\sqrt{{\left(C_{1}-C_{2}\right)}^{2}+{\left(\gamma_{1}-\gamma_{2}\right)}^{2}}$$

In the traditional ABC algorithm, the formula for generating a new food source is:(6)$$v_{\mathrm{ij}}=x_{\mathrm{ij}}+\emptyset_{\mathrm{ij}}\left(x_{\mathrm{ij}}-x_{\mathrm{ij}}\right)$$

The formula for the selection of food sources by scout bees are as follows:(7)$$P_{i}=\frac{{\mathrm{fit}}_{i}}{\sum_{n=1}^{\mathrm{SN}}{\mathrm{fit}}_{n}}\;$$
(8)$${\mathrm{fit}}_{i}=\{\begin{array}{c} \frac{1}{1+{\mathrm{fit}}_{i}},\;{\mathrm{fit}}_{i}\geq 0\\ 1+\mathrm{abs}\left({\mathrm{fit}}_{i}\right),\;{\mathrm{fit}}_{i}<0 \end{array}\;$$
where j∈[1,2,⋯,D] and k∈[1,2,⋯,SN] are selected randomly. $\emptyset_{\mathrm{ij}}\in \left[-1,1\right]$, which denotes a random value; ${\mathrm{fit}}_{i}$ is the fitness value corresponding to the food source. When the value of $\emptyset_{\mathrm{ij}}$ is small, the search range of the scout bees is small, which causes the algorithm to not converge or to converge in advance; On the contrary, the optimal solution may be ignored, thus affecting the convergence of the algorithm. Therefore, this article attempts to improve the convergence of the ABC algorithm.

Third, the weight value is defined as ${∆}_{i}=\frac{d_{i}}{d_{\max}}$, in the range of (0, 1), where the value of $d_{i}$ is the distance between the current solution and the optimal solution, and the value of $d_{\max}$ is obtained by substituting the vertex (1, 0) and the vertex (100, 0.1) into Equation (1). The range of the food-source-update solution can be adjusted automatically by ${∆}_{i}$. If ${∆}_{i}$ is smaller, it means that the search range of the update solution is smaller. Otherwise, it is larger. This update strategy can effectively reduce the number of iterations. Plugging ${∆}_{i}$ into Equation (6) gives the new formula for improved food sources, as follows.
(9)$$v_{\mathrm{ij}}=x_{\mathrm{ij}}+{∆}_{i}\emptyset_{\mathrm{ij}}\left(x_{\mathrm{ij}}-x_{\mathrm{ij}}\right)\;$$

In view of the lack of mathematical theory to guide the parameter search of support vector machines, the low efficiency of traditional artificial bee colony algorithm (ABC) search and the tendency to generate local optimal solutions, an artificial bee colony support vector machine model (IABC-SVM) based on a chaotic-update strategy is proposed. The model improves the convergence speed and classification accuracy of the ABC algorithm through two-dimensional uniform population initialization and food source update based on Euclidean distance.

## 3. Results and Discussion

In order to assess the performance of the proposed IABC-SVM model, experimental results obtained with the IABC-SVM are compared with the gray wolf algorithm optimization support vector machine (GWO-SVM) [27], the gravity search algorithm optimization support vector machine (GSA-SVM) [28] and the ABC-SVM model. For fair comparison, all of these algorithms were run with the same parameters. A comparative experiment was conducted on the UCI data set to illustrate the effectiveness and feasibility of the improved ABC algorithm in the optimal selection parameters (c,g).

### 3.1. IABC-SVM Algorithm Test

In order to prove the robustness and performance of the IABC, results have been compared with those obtained by using other optimization algorithms, such as ABC, GSA and GWO. Through multiple trainings, the number of iterations and classification accuracy of each model are observed when they converge.

Figure 3 depicts the convergence characteristics of the chaotic-based ABC-SVM model and the aforementioned optimization algorithms. It can be seen that the proposed classification model is clearly faster than the other models, since it reaches the accepted optimum precision and number of iterations. After several iterations, the maximum classification accuracy of the ABC-SVM and IABC-SVM model can reach 100%. However, the convergence speed of the IABC-SVM model is significantly faster than that of the ABC-SVM model. The accuracy of the GSA-SVM and GWO-SVM models has been fluctuating up and down. No convergence can be achieved. Among them, the accuracy of the GSA-SVM model has been hovering around 40%, and the accuracy of the GWO-SVM model fluctuates greatly, with a maximum of 100% and a minimum of 73%. It can be seen that the IABC-SVM model could preclude bees from being trapped in the local optimum. In addition, the accuracy of the IABC-SVM model is always 100%, which allowed us to conclude that the IABC-SVM model was more robust and more accurate.

### 3.2. SVM Model Based on Spectra Preprocessing

A total 240 samples were divided into training and test sets. The training set consisted of 180 rice samples (45 samples were selected from resistant Longjing 34, high-resistance Longjing 33, susceptible Longjing 56, high-susceptibility Longjing 36). The test set consisted of the remaining 60 rice samples; 15 samples were selected from each of the four differently resistant rice seeds.

The IABC-SVM and ABC-SVM algorithm parameter settings are the same, with the total number of bees being 30, a bee number of 15, a maximum search number of 100 and the parameters of 2. When the maximum number of iterations is reached, the algorithms are stopped. The parameters (c,g) in the four models have the same value range, [0, 100] and [0, 0.01], respectively. Then, the ABC-SVM, IABC-SVM, GSA-SVM and GWO-SVM models were established after using different preprocessing data and the original spectral data, respectively. The differently-resistant-rice-classification models of ABC-SVM, IABC-SVM, GSA-SVM and GWO-SVM were established as follows, in Table 2, Table 3, Table 4 and Table 5.

The Raw-ABC-SVM, Raw-IABC-SVM, Raw-GSA-SVM and Raw-GWO-SVM rice-classification models based on the full spectrum were established to have accuracy of 60%, 71%, 71% and 73%, respectively. Comparing the five preprocessing methods, it has been found that the modeling results of SNV is better than that of SGD, SGS, Base and MSC. After SNV, the ABC-SVM, GSA-SVM and GWO-SVM models have greatly improved accuracy, and the SNV-ABC-SVM model can reach up to 91% accuracy. As can be seen, the accuracy of models built by preprocessing was improved by different degrees. The SNV-IABC-SVM model had better precision, and the running time was reduced to 13 s. Similarly, none of the indices of the other four preprocessing IABC-SVM models outperformed the SNV-IABC-SVM model. In a word, the “SNV-IABC-SVM” modeling methods can fully realize the fast and accurate classification of the rice seeds with different resistance used in the experiment.

### 3.3. SVM Model Based on Feature Extraction

The volume of the raw spectral data is large, and the modeling process is complicated, so the random forest algorithm was used to extract characteristic variables to improve modeling performance and reduce running time in this study. After RF, the ABC-SVM, IABC-SVM, GSA-SVM and GWO-SVM were established as follows, in Table 6, Table 7, Table 8 and Table 9.

The RF-ABCSVM, RF-IABCSVM, RF-GSASVM and RF-GWOSVM rice-classification models were established to have accuracy of 70%, 70%, 46% and 70%, respectively. The IABC-SVM is greatly improved in speed and precision compared with the ABC-SVM, GSA-SVM and GWO-SVM. The IABC-SVM exhibited higher precision and faster speed than the ABC-SVM in classification. The performance of the RF-SNV-IABCSVM model has been greatly improved due to the precision increasing by 8%, 7% and 55%, respectively, and due to the running time declining by 10 s, 1 s and 1 s, respectively, compared with the RF-SNV-ABCSVM, RF-SNV-GWOSVM and RF-SNV-GSASVM models. The GSA-SVM model has a relatively poor classification ability. The SNV can effectively improve the accuracy of the model, and RF could reduce scale of the original spectrum data. As can be seen from the tables, the RF-SNV-IABCSVM model has better fitting performance and more reliable accuracy in classification compared with the other three models.

### 3.4. Seed-Classification Evaluation Experiment

In order to facilitate seed selection, we conducted a seed-evaluation experiment. In this experiment, a resistance–susceptibility classification model was established, with 240 spectra divided in which two categories, namely, resistance (resistant Longjing 34 and high-resistance Longjing 33) and susceptibility (susceptible Longjing 36 and high-susceptibility Longjing 56). With 75% of the samples as a training set and 25% as a test set, SVM, ABC-SVM, IABC-SVM, GSA-SVM and GWO-SVM optimal models were built to test the classification ability of RF-SNV-IABCSVM hybrid model. The specific results are shown in Table 10.

According to the Technical Specification for Disease Resistance of Rice Varieties (http://www.docin.com/p-255618735.html accessed on 30 May 2020), rice resistance is graded from high to low in order of high resistance, resistance, medium resistance, susceptible, medium susceptible and high susceptible. Thus, a total of 300 spectral data (150 of medium-resistance and 150 of medium-susceptibility) were scanned by the Ranman spectrometer. The optimal models of SVM, ABC-SVM, IABC-SVM, GSA-SVM and GWO-SVM were established as follows, in Table 11.

The experimental results show that the classification ability of the RF-SNV-IABCSVM hybrid model is better than the single RF-SNV-ABCSVM and SNV-IABC-SVM model and that RF-SNV-IABCSVM can make full use of the performance of the RF and IABC algorithms. Therefore, it is feasible to use Ranman spectroscopy combined with stoichiometry to quickly identify rice seeds with different resistance, which provides a new method for the rapid detection of rice resistance.

## 4. Conclusions

In order to study the classification of rice-blast-resistant seeds, the Ranman spectra of 14 rice varieties were scanned. Five different pretreatment methods were used. the random forest algorithm was used to extract characteristic variables, reduce the correlation between variables, shorten the running time and improve the accuracy of classification model. A resistant-rice-classification model based on the IABC-SVM optimization algorithm is proposed. Improving the traditional bee colony algorithm accelerated the convergence speed to obtain a global optimal solution. By optimizing the parameters of SVM, the generalization and accuracy of SVM classification are improved. Finally, by comparing the IABC-SVM, ABC-SVM, GSA-SVM and GMO-SVM models, the accuracy of IABC-SVM increases up to 100%, while the accuracy of ABC-SVM, GSA-SVM and GMO-SVM increases to 93%, 50% and 93%, respectively. Based on the result of the experiments mentioned above, the model IABC-SVM not only has a fast running speed but also demonstrates good performance in the classification of the different rice seeds. This classification experiment can provide evidence for the further development of Ranman spectroscopy in the detection of other crop seeds.

## Figures and Tables

**Figure 1 molecules-27-04091-f001:**
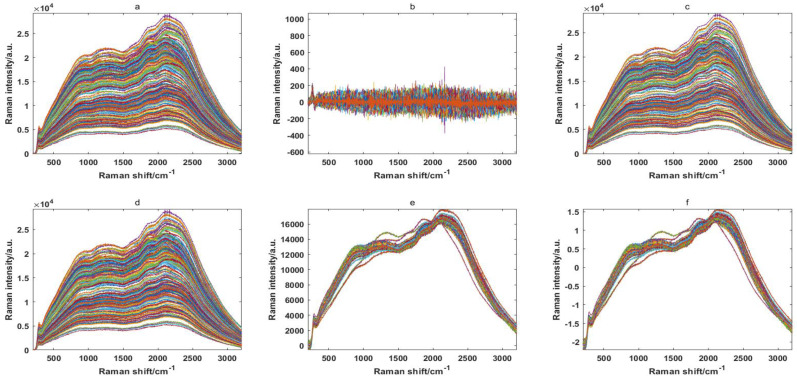
Original Ranman spectra (**a**) and Ranman spectra modified by removing the noise signal for 240 rice seed samples using Savitzky–Golay 1-Der (**b**), Savitzky–Golay smoothing (**c**), baseline (**d**), multivariable scatter correction (**e**) and standard normal variable (**f**).

**Figure 2 molecules-27-04091-f002:**
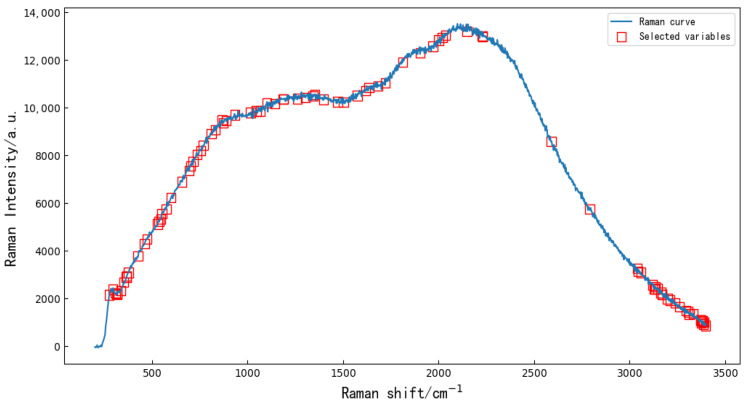
Random forest algorithm was used to extract 90 characteristic variables from the original Ranman spectra.

**Figure 3 molecules-27-04091-f003:**
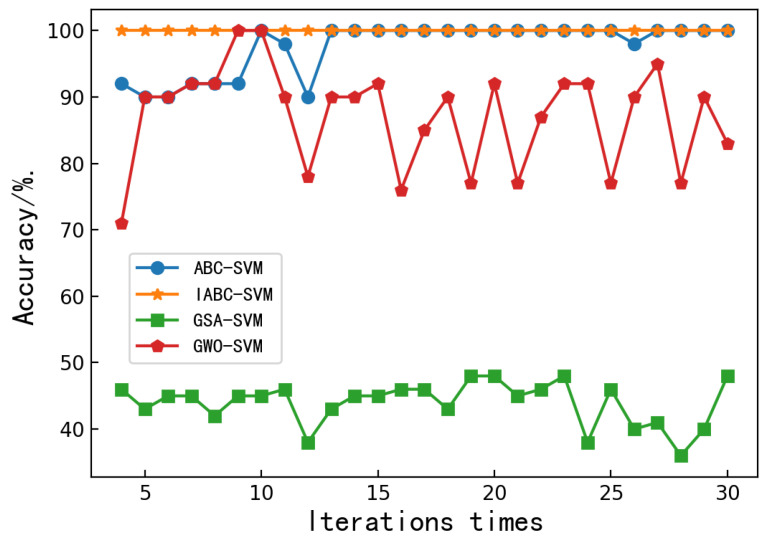
Relationship between classification accuracy and iteration times of the ABC-SVM, IABC-SVM, GSA-SVM and GWO-SVM models.

**Table 1 molecules-27-04091-t001:** Support vector machine, BP neural network and probabilistic neural network resistant-rice-classification models established using original Ranman spectra.

Model	Input Units	Time (s)	Accuracy (%)
Raw − SVM	3202	3	45
Raw + BP	3202	487	50
Raw + PNN	3202	2	25

**Table 2 molecules-27-04091-t002:** The ABC-SVM classification models established by the original Ranman spectra and the Ranman spectra after five different kinds of pretreatments.

Model	Misjudgment (Train/Test)	Time (s)	Train (%)	Test (%)
Raw-ABC-SVM	0/39	28	100	60
SNV-ABC-SVM	0/5	28	100	91
MSC-ABC-SVM	0/5	31	100	91
BASE-ABC-SVM	0/35	33	100	41
SGS-ABC-SVM	0/19	29	100	68
SGD-ABC-SVM	0/45	34	100	25

**Table 3 molecules-27-04091-t003:** The IABC-SVM classification models established by the original Ranman spectra and the Ranman spectra after after five different kinds of pretreatments.

Model	Misjudgment (Train/Test)	Time (s)	Train (%)	Test (%)
Raw-IABC-SVM	0/17	12	100	71
SNV-IABC-SVM	0/0	13	100	100
MSC-IABC-SVM	0/0	15	100	100
BASE-IABC-SVM	0/16	15	100	73
SGS-IABC-SVM	0/17	15	100	71
SGD-IABC-SVM	0/35	18	100	41

**Table 4 molecules-27-04091-t004:** The GSA-SVM classification models established by the original Ranman spectra and the Ranman spectra after after five different kinds of pretreatments.

Model	Misjudgment (Train/Test)	Time (s)	Train (%)	Test (%)
Raw-GSA-SVM	0/45	36	100	25
SNV-GSA-SVM	0/33	35	100	45
MSC-GSA-SVM	0/36	16	100	40
BASE-GSA-SVM	0/45	16	100	25
SGS-GSA-SVM	0/45	16	100	25
SGD-GSA-SVM	0/45	16	100	25

**Table 5 molecules-27-04091-t005:** The GWO-SVM classification models established by the original Ranman spectra and the Ranman spectra after after five different kinds of pretreatments.

Model	Misjudgment (Train/Test)	Time (s)	Train (%)	Test (%)
Raw-GWO-SVM	0/16	2	100	73
SNV-GWO-SVM	0/7	24	100	88
MSC-GWO-SVM	0/9	21	100	85
BASE-GWO-SVM	0/35	16	100	41
SGS-GWO-SVM	0/45	17	100	25
SGD-GWO-SVM	0/37	16	100	38

**Table 6 molecules-27-04091-t006:** The Ranman spectra after five different pretreatments are extracted using the random forest algorithm, and the ABC-SVM classification models are established using the extracted feature variables.

Model	Misjudgment (Train/Test)	Time (s)	Train (%)	Test (%)
RF-ABCSVM	0/18	18	100	70
RF-SNV-ABCSVM	0/5	19	100	92
RF-MSC-ABCSVM	0/4	14	100	93
RF-Base-ABCSVM	0/19	12	100	68
RF-SGS-ABCSVM	0/15	12	100	75
RF-SGd-ABCSVM	0/37	14	100	33

**Table 7 molecules-27-04091-t007:** The Ranman spectra after five different pretreatments are extracted using the random forest algorithm, and the IABC-SVM classification models are established using the extracted feature variables.

Model	Misjudgment (Train/Test)	Time (s)	Train (%)	Test (%)
RF-IABCSVM	0/18	9	100	70
RF-SNV-IABCSVM	0/0	9	100	100
RF-MSC-IABCSVM	0/4	7	100	93
RF-Base-IABCSVM	0/16	6	100	73
RF-SGS-IABCSVM	0/15	6	100	75
RF-SGd-IABCSVM	0/37	8	100	33

**Table 8 molecules-27-04091-t008:** The Ranman spectra after five different pretreatments are extracted using the random forest algorithm, and the GSA-SVM classification models are established using the extracted feature variables.

Model	Misjudgment (Train/Test)	Time (s)	Train (%)	Test (%)
RF-GSASVM	0/32	10	100	46
RF-SNV-GSASVM	0/33	10	100	45
RF-MSC-GSASVM	0/30	8	100	50
RF-Base-GSASVM	0/29	8	100	52
RF-SGS-GSASVM	0/28	8	100	53
RF-SGd-GSASVM	0/45	7	100	25

**Table 9 molecules-27-04091-t009:** The Ranman spectra after five different pretreatments are extracted using the random forest algorithm, and the GWO-SVM classification models are established using the extracted feature variables.

Model	Misjudgment (Train/Test)	Time (s)	Train (%)	Test (%)
RF-GWOSVM	0/45	10	100	70
RF-SNV-GWOSVM	0/4	10	100	93
RF-MSC-GWOSVM	0/5	7	100	92
RF-Base-GWOSVM	0/46	8	100	23
RF-SGS-GWOSVM	0/45	7	100	25
RF-SGd-GWOSVM	0/39	7	100	35

**Table 10 molecules-27-04091-t010:** After SNV preprocessing, the random forest algorithm is used for feature extraction, and the resistance–susceptibility classification models are established by four optimized support vector machine algorithms.

Model	Misjudgment (Train/Test)	Time (s)	Train (%)	Test (%)
RF-SNV-SVM	0/27	3	100	55
RF-SNV-ABCSVM	0/0	17	100	100
RF-SNV-IABCSVM	0/0	8	100	100
RF-SNV-GSASVM	0/14	9	100	77
RF-SNV-GWOSVM	0/1	9	100	98

**Table 11 molecules-27-04091-t011:** After SNV preprocessing, the random forest algorithm is used for feature extraction, and the actual breeding resistance–susceptibility classification models established by four optimized support vector machine algorithms.

Model	Misjudgment (Train/Test)	Time (s)	Train (%)	Test (%)
RF-SNV-SVM	0/40	5	100	75
RF-SNV-ABCSVM	0/3	59	100	98
RF-SNV-IABCSVM	0/0	17	100	100
RF-SNV-GSASVM	0/30	30	100	81
RF-SNV-GWOSVM	0/18	23	100	89

## Data Availability

Not applicable.

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
