# Peer review of "The Classification of Rice Blast Resistant Seed Based on Ranman Spectroscopy and SVM"

_molecules, 2022, doi:10.3390/molecules27134091_

Round 1
Reviewer 1 Report
The authors present a study on the coupling of Raman spectroscopy and Machine Learning to classify Rice blast resistant rice seeds.
From spectroscopic results, some support vector machine models were established. The results obtained are promising.
It is recommended that the following adjustments be made to the document.
1. The title does not adequately represent the content of the manuscript. the topic of support vector machine models should be introduced.
2. Include in the keywords, some term related to support vector machine models
3. It is advised to adjust the introduction presented the advantages of the raman spectroscopy, its potential in research area. In addition, the topic of support vector machine models should be introduced in a more forceful way.
Author Response
Response to Reviewer 1 Comments
Point 1: The title does not adequately represent the content of the manuscript. the topic of support vector machine models should be introduced.
Response 1: The title has been changed to The classification of rice blast resistant seed based on Ranman spectroscopy and SVM.
Point 2: Include in the keywords, some term related to support vector machine models
Response 2: Artificial bee colony algorithm related to support vector machine is added to the keyword.
Point 3: It is advised to adjust the introduction presented the advantages of the raman spectroscopy, its potential in research area. In addition, the topic of support vector machine models should be introduced in a more forceful way.
Response 3: The presentation has been reorganized; support vector machine related studies have been added and presented the advantages of the raman spectroscopy.

Reviewer 2 Report
The authors proposed a method to classify the rice blast resistant seed based on Raman spectroscopy. It is very interesting and it could provide a new tool to in-field detection. However, some minor midifications should be done before it could be accepted.
1. Raman should always be in the capital form.
2. The authors used the laser of 532 nm, have you ever tried any other nm? such as 785 to avoid the fluorescence.
3. In your research, you have 4 different kinds of seeds. Have you ever try to use another brand of seed to verify your model. Because it is very important if you want to use it in practice.
4. Details should be given about the Raman spectrometer such as type and company.
5. Figures with higher resolution should be provided in the manuscript.
Author Response
Response to Reviewer 2 Comments
Point 1: Raman should always be in the capital form.
Response 1: Raman has been changed to uppercase.
Point 2: The authors used the laser of 532 nm, have you ever tried any other nm? such as 785 to avoid the fluorescence.
Response 2: At present, the instrument only has a laser of 532 nm. In the future, I want to try to buy it and try it out.
Point 3: In your research, you have 4 different kinds of seeds. Have you ever try to use another brand of seed to verify your model. Because it is very important if you want to use it in practice.
Response 3: In Section 3.4 of the article, a total of 10 varieties of medium resistance and medium susceptible were added to the previous four varieties for new experiments to test the universality of the model.
Point 4: Details should be given about the Raman spectrometer such as type and company.
Response 3: The spectrometer details that have been perfected are the Advantage 532 benchtop Ranman spectrometer produced by DeltaNu, USA.
Point 5: Figures with higher resolution should be provided in the manuscript.
Response 5: All have been modified to high-resolution pictures.